# Macro-Determinants of NEET: An Ecological Study at the Country Level of Analysis for the Period 1997–2020

Simone Amendola 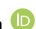

Department of Dynamic and Clinical Psychology, and Health Studies, Faculty of Medicine and Psychology, Sapienza University of Rome, 00185 Rome, Italy; simone.amendola@uniroma1.it

**Abstract:** The aim of the present study was to examine macro-determinants of the Not Engaged in Employment, Education or Training (NEET) rate with the country as the unit of analysis. Data from 40 countries were extracted from the Organisation for Economic Co-operation and Development (OECD) database. Linear mixed models were used to examine associations between the NEET rate and gross domestic product (GDP), population density, life expectancy, social spending, homicide rate, part-time employment, poverty, social inequality (GINI index), and education spending. As part of a sensitivity analysis, the analysis was repeated using open data from the World Bank Group. GDP and social spending were uniquely associated with the NEET rate after controlling for the effects of other factors. Social inequality, poverty, and education spending showed borderline significant associations with the NEET rate. The findings of the present ecological study showed associations between environmentally unfavourable conditions or harshness and the NEET rate at the country level and may inform appropriate policy measures to contain and promote a decrease in the NEET rate.

**Keywords:** NEET rate; youth unemployment; risk of marginalization; trends





## 1. Introduction

At least one in ten European adolescents and emerging adults (15–29 years old) followed unsuccessful pathways of withdrawal from both the labour market and education in 2021, the so-called "Not Engaged in Employment, Education or Training" (NEET) rate, ranging from 5.5% in the Netherlands to 23.1% in Italy [1]. NEET status is explained not only by individual choices and characteristics (e.g., female gender, school attainment) but also by family background characteristics (e.g., parental education and jobless condition, sibling labour force status, immigrant parents) and socioeconomic factors [2–11]. Despite the NEET concept itself being somewhat problematic (e.g., it refers to a heterogeneous group of people, it defines young people in negative terms highlighting what they are not, it is associated with negative connotations), the NEET phenomenon guarantees increasing attention from mental and public health experts [11–19]. Long-term (more than five months) NEET status is associated with risk factors, such as taking a career role, seclusion at home, involvement in deviant activities, and addiction or health problems [11]. Inactivity and discomfort and a sense of uncertainty and instability are associated with the NEET condition [15]. Peer problems and symptoms of psychopathology generally precede the NEET condition [12,13,16–18] despite the fact that the inverse path has also been demonstrated [14]. A previous study [19] examined the risk of becoming socially and economically marginalized in Japan, showing that more than half of NEET participants (1.86% of the total sample) were in a hikikomori condition (1.06%). Accordingly, NEET and hikikomori could share psychological tendencies related to occupational and social withdrawal.

The risk of marginalization is associated with socioeconomic and cultural changes such as economic recession, job insecurity, and globalization [20–24]. It has been proposed that the existence of youth withdrawing from social and/or occupational life may represent a feasible life strategy associated with high-resource-abundance societies that would be

maladaptive in poor societies [25]. In accordance, the NEET rate would positively correlate with the gross domestic product (GDP) per capita and social assistance policies at the population level of analysis [25]. To explain the association between resource availability and the risk of marginalization, some authors [25] have referred to the Life History Theory [26,27]. It posits that organisms pursue strategies that best help them allocate energy towards different survival and reproductive activities in a way that maximises their fitness [28]. Ecology unpredictability and harshness (e.g., levels of morbidity and mortality, resource availability, population density, intrasexual competition, and predator threat) are major factors in determining the optimal strategy of energy allocation [25,28–32].

Few studies have examined the macro-determinants of NEET. Bruno et al. [33] analysed the association between GDP and the NEET rate in European regions between 2000 and 2010. Bingol [34] explored the associations between GDP, inflation rate, education expenditure, foreign direct investment, human development index data, and the NEET rate in six countries (i.e., Russia, Brazil, India, Indonesia, South Africa, and Turkey) between 2005 and 2018. Caroleo et al. [35] analysed the different macro-determinants (i.e., school-to-work transition, labour market, and institutional variables) of the NEET status in 21 European countries in two selected years, 2007 and 2016. These studies showed negative associations between the NEET rate and GDP (i.e., as GDP increases, the NEET rate decreases) [33–35], education expenditure (i.e., as education expenditure increases, the NEET rate decreases) [34,35], and active labour market policies [35]. Further, positive associations between the NEET rate and the human development index [34] and passive subsidies to the unemployed were demonstrated [35]. On the contrary, another study [8] highlighted a positive correlation between GDP and the NEET rate. However, the findings of these studies were limited by the analysis of data from only a few countries, by the brief time periods considered, and/or by a small number of factors examined.

In light of the above, the main aim of the present study was to examine the associations between the NEET rate, GDP per capita, and social expenditure between 1997 and 2020 considering European and non-European countries. The hypothesis that rates of NEET would be positively correlated with GDP per capita and social expenditure (i.e., the NEET rate would be a sign of wealthy ecologies) was tested with the country as the unit of analysis. Further, the associations between the NEET rate and factors related to ecologically unfavourable conditions or unpredictability and harshness [25–32], such as population density, life expectancy, and violent crime (i.e., homicide rate), were explored. The role of poverty, social inequality (GINI index), education spending, and the percentage of part-time employment (as a proxy for job insecurity and environmental unpredictability [36]) was also investigated.

## 2. Materials and Methods

### 2.1. Data Source and Variables

All data were extracted from the Organisation for Economic Co-operation and Development (OECD) database [37] on 18 June 2022. The definitions of the variables considered for analysis are summarized in Table S1 (Supplementary Material). The study was exempt from approval by an ethics committee because all data were collected in aggregated format from publicly available databases.

### 2.2. Statistical Analysis

Data from 40 countries were considered for analysis due to the availability of NEET data for at least one year for the period 1997–2020. The following analysis was conducted applying listwise deletion on the available data for a total of 767 potential observations for which data on NEET were available. To explore the associations between the NEET rate and the variables of interest, linear mixed models were fitted, including random (country-specific) intercept and modelling first-order autoregressive covariance for the residuals within the country using the restricted maximum likelihood estimates of parameters. To test the study hypothesis, two models were fitted analysing associations between the NEET rate

and GDP, population density, life expectancy, and social spending (model 1), including the poverty rate, education spending, income inequality, the homicide rate, and the percentage of part-time employment (model 2). The analysis was repeated using open data from the World Bank Group [38] (extracted on 27 June 2022) as part of a sensitivity analysis. All analyses were carried out using R version 3.3.0.

## 3. Results

NEET rate trends according to each country for the period 1997–2020 are displayed in Figure 1, whereas descriptive characteristics and bivariate associations are reported in Tables S2 and S3 (Supplementary Material).

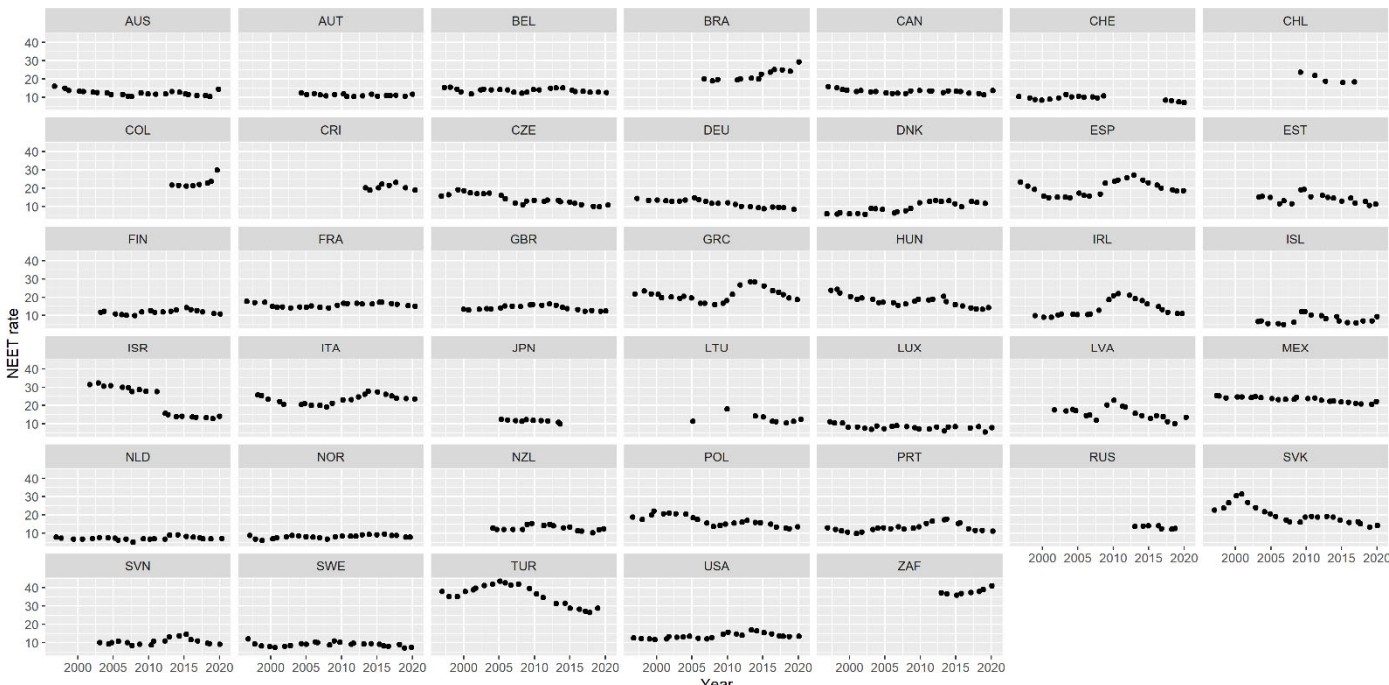

**Figure 1.** National trends in NEET rate for the 40 countries considered for analysis, 1997–2020 (OECD data). *Note*. AUS: Australia, AUT: Austria, BEL: Belgium, BRA: Brazil, CAN: Canada, CHE: Switzerland, CHL: Chile, COL: Colombia, CRI: Costa Rica, CZE: Czech Republic, DEU: Germany, DNK: Denmark, ESP: Spain, EST: Estonia, FIN: Finland, FRA: France, GBR: United Kingdom, GRC: Greece, HUN: Hungary, IRL: Ireland, ISL: Island, ISR: Israel, ITA: Italy, JPN: Japan, LTU: Lithuania, LUX: Luxembourg, LVA: Latvia, MEX: Mexico, NLD: the Netherlands, NOR: Norway, NZL: New Zealand, POL: Poland, PRT: Portugal, RUS: Russia, SVK: Slovakia, SVN: Slovenia, SWE: Sweden, TUR: Turkey, USA: United States, ZAF: South Africa.

GDP and social spending were uniquely associated with the NEET rate after controlling for the effects of other variables (model 1, Table 1). Their effects persisted after controlling for important covariates (model 2, Table 1). Among the latter, social inequality, poverty, and education spending showed borderline significant associations with the NEET rate (*p*-values = 0.057, 0.09, and 0.10, respectively).

The sensitivity analysis, as reported in Tables S4 and S5 (Supplementary Material), provides further evidence of the associations between the NEET rate, GDP, and social inequality, whereas the association between the NEET rate and social spending was not statistically significant when controlling for the covariates.

**Table 1.** Results of mixed models predicting associations between the NEET rate and the variables of interest, 1997–2020 (OECD data).

| | Model 1 [a] | | | | | Model 2 [b] | | | | |
|---|---|---|---|---|---|---|---|---|---|---|
| | β (SE) | 95%CI | t (df = 234) | B | *p* | β (SE) | 95%CI | t (df = 73) | B | *p* |
| Intercept | 0.02 (0.15) | −0.28, 0.32 | 0.14 | 15.48 | 0.89 | −0.07 (0.15) | −0.37, 0.23 | −0.48 | 16.64 | 0.63 |
| Time | 0.09 (0.08) | −0.07, 0.25 | 1.12 | 0.10 | 0.26 | −0.001 (0.08) | −0.17, 0.16 | −0.01 | −0.00 | 0.99 |
| Gross domestic product (per capita) | −0.47 (0.11) | −0.69, −0.25 | −4.21 | −0.19 | <0.001 | −0.44 (0.15) | −0.75, −0.13 | −2.85 | −0.16 | 0.006 |
| Population density | −0.04 (0.16) | −0.36, 0.27 | −0.26 | −0.00 | 0.79 | −0.08 (0.14) | −0.37, 0.21 | −0.56 | −0.01 | 0.57 |
| Life expectancy | −0.02 (0.11) | −0.24, 0.2 | −0.20 | −0.06 | 0.84 | 0.02 (0.17) | −0.32, 0.36 | 0.10 | 0.04 | 0.92 |
| Social spending | 0.20 (0.07) | 0.05, 0.34 | 2.61 | 0.24 | 0.01 | 0.28 (0.14) | 0.01, 0.55 | 2.09 | 0.36 | 0.04 |
| Homicide | – | – | – | – | – | 0.02 (0.10) | −0.19, 0.22 | 0.15 | 0.1 | 0.88 |
| Part-time job | – | – | – | – | – | −0.03 (0.16) | −0.35, 0.29 | −0.18 | −0.02 | 0.86 |
| Poverty | – | – | – | – | – | 0.30 (0.18) | −0.06, 0.66 | 1.66 | 0.46 | 0.10 |
| Social inequality | – | – | – | – | – | 0.38 (0.2) | −0.01, 0.77 | 1.94 | 40.81 | 0.06 |
| Education spending | – | – | – | – | – | −0.16 (0.09) | −0.34, 0.03 | −1.72 | −1.27 | 0.09 |

[a]: all variables fitted jointly except covariates (274 observations for 35 countries), [b]: all variables fitted jointly including covariates (111 observations for 28 countries), β: standardized coefficient, SE: standard errors, CI: confidence interval, t: value of the t statistic, df: degree of freedom, B: unstandardized coefficient. All predictors were centred on the mean.

## 4. Discussion

The present ecological study examined macro-determinants of the NEET rate at the country level. The results of principal and sensitivity analyses consistently showed a negative association between national GDP (per capita) and the NEET rate; i.e., as GDP increases, the NEET rate decreases. These results are in line with those of recent research demonstrating significant associations between these variables both in fragile economies [34] and in European countries [33]. In a previous study [35], national GDP per capita exerted a significant effect on the NEET rate even after controlling for personal characteristics predisposed to NEET status (e.g., gender, parents' education, marital status, disabling conditions, immigrant status). The findings of the present study expand those of previous research, demonstrating a significant negative association between GDP and the NEET rate in both European and non-European countries over a long period of time, i.e., between 1997–2020. This result suggests that appropriate policy measures focused on socioeconomic factors and increasing investments and country-level productivity may contain and promote a decrease in the NEET rate [35]. Further, the results of the statistical analysis showed a positive association between social expenditure and the NEET rate. A previous study [35] demonstrated a positive association between passive subsidies to the unemployed and the NEET rate. Taken together, the study findings provide only partial support to the study hypothesis (i.e., the NEET rate as a sign of wealthy ecologies). Despite a positive association between social expenditure and the NEET rate, GDP per capita was not positively associated with the NEET rate; rather, it was negatively associated.

Further, some aspects indicating environmental harshness or national unfavourable conditions, such as social inequality as measured by the GINI index and the poverty rate, correlated positively with the NEET rate at the population level of analysis. These findings broaden scientific knowledge on the socioeconomic determinants of NEET, indicating that as a country's social inequality and poverty increase, the NEET rate increases. Importantly, a tendency towards a significant negative association between education spending and the NEET rate was also observed. That is, as education spending increases, the NEET rate tends to decrease. Taken together, these findings support and expand those of other recent studies demonstrating that the share of NEET may derive from objective disadvantageous conditions at the country level. [35,39]. Consequently, active labour market programmes, apprenticeships, vocational education, and training programmes are useful measures that could be implemented to favour the school-to-work transition and youth employment [35,39,40].

The results of the present study need to be interpreted considering some limitations. First, the analyses are based on national and not individual data. Considering that the term NEET refers to a heterogeneous group of individuals, the analysis of individual data is

needed to further explore the study hypothesis (e.g., using multi-level analysis) and draw firm conclusions at the individual unit of analysis [9]. A second limitation concerns the fact that the study results depend on primary data quality (i.e., data reporting and collection procedures). Despite sensitivity analyses being performed to minimize reliability problems, their results should be interpreted with caution due to differences in the operational definitions of some variables (e.g., NEET age coverage, social and education spending) between the two databases used for principal and sensitivity analysis (see Supplementary Material for additional information). Finally, labour force surveys were the preferred source of data on the NEET rate, and employment was generally defined according to a specific brief period (i.e., the reference week of the survey) (see Supplementary Material).

In conclusion, the findings of the present study highlighted that the NEET rate may result from unfavourable conditions at the country level of analysis. They emphasize the need to increase economic growth to reduce NEET rates. Social spending, social inequality, and poverty positively correlated with the NEET rate, whereas the increase in education spending may represent a protective factor at the country unit of analysis. Thus, decreasing inequality and improving economic opportunities and education spending (e.g., school-to-work transition programs) may help in containing and reducing the NEET rate.

**Supplementary Materials:** The following supporting information can be downloaded at: https: //www.mdpi.com/article/10.3390/youth2030028/s1, Table S1: Variables included in the principal analysis and their definition as extracted from the OECD database; Table S2: Descriptive characteristics (mean and standard deviation) of the analysed OECD data for the period of 1997–2020; Table S3: Results of mixed models predicting bivariate associations between the NEET rate and variables of interest in countries considered for analysis controlling for the effect of time, 1997–2020 (OECD data); Table S4: Results of mixed models predicting bivariate associations between the NEET rate and variables of interest in countries considered for analysis controlling for the effect of time (World Bank data restricted to 40 countries); Table S5: Results of mixed models predicting associations between the NEET rate and variables of interest (World Bank data restricted to 40 countries); Table S6: Results of mixed models predicting bivariate associations between the NEET rate and variables of interest in countries considered for analysis controlling for the effect of time (World Bank data); Table S7: Results of mixed models predicting associations between the NEET rate and variables of interest (World Bank data); Table S8: Countries data analysed per year in model 2; Table S9: Country data analysed per country in model 2; Figure S1: National trends in the NEET rate for the 40 countries considered for sensitivity analysis, 1997–2020 (World Bank data); Figure S2: National trends in the NEET rate for the 167 countries considered for sensitivity analysis, 1997–2020 (World Bank data) (including countries in supplementary material).

**Funding:** This research received no external funding.

**Institutional Review Board Statement:** Not applicable.

**Informed Consent Statement:** Not applicable.

**Data Availability Statement:** The databases from which the data were extracted are publicly available online (Organisation for Economic Co-operation and Development (OECD) database and open data of the World Bank Group).

**Conflicts of Interest:** The author declares no conflict of interest.

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
