# Peer review of "Macro-Determinants of NEET: An Ecological Study at the Country Level of Analysis for the Period 1997–2020"

_2673-995X, doi:10.3390/youth2030028_

Round 1
Reviewer 1 Report
General comments:
Thank you for opportunity for reviewing this interesting paper. The topic of the manuscript is interesting.
I believe that this manuscript doesn´t qualify for acceptance at this time and should be improved for publication.
Specific comments:
1. Writing
I recommend an English proofreading of the whole text, as some phrases should be corrected, for example in the abstract: “The present study aimed at examining macro-determinants of Not engaged in Employment, Education or Training (NEET) rate at the nation unit of analysis”.
2. Title
The title does not fully reflect the content and problem studied. Please consider using the “NEET” phrase in the title, if the study has not only considered exclusion of youth from education and work, but also from trainings (apprenticeship, VET trainings, etc.).
3. Abstract
The abstract reflects the manuscript and provide a summary of what was done and what was found.
4. Key Words
The keywords are representative of the subject of the study. I recommend replacement of “marginalization” with “risk of marginalization” or removing the keyword.
5. Introduction
The introduction reflects the state of the art in relation to the study. What is missing is mentioning criticisms linking the conceptualization of NEETs. Please see: DOI: 10.1080/02673843.2022.2065922, DOI: 10.7862/rz.2018.hss.77.
6. Materials and Methods
The definition of NEET applied is the weakest point. There is provided the age range of young people (15-29), but there is not mentioned the minimum time being Not in Employment, Education, and Training status, as depending on the source, young person may be included in the NEET group for as little as 4 weeks of inactivity (European Union Employment Committee definition), 3 or 6 or 12 or even 24 or more months. The above is also missing in the NEET definition in Table S1. Does OECD used the same definition of NEET as the World Bank Group in the database? Have they applied the same minimal inactivity period? Please improve.
There are provided only general references to OECD and WB databases: https://stats.oecd.org/ , https://data.worldbank.org/. However, it is not mentioned which indicator or indicators were used for the purpose of analyses or to determine the NEET variable. Please improve.
Please also confirm that the analyses included NEET group in 3 aspects, as excluded from employment, education, but also from trainings, as the title refers only to the first two activities. Person not working and not involved in a school or an university education, but having, for example, VET courses, trainings, internships should not be included in the NEET category.
Moreover, the definition of NEET (in Table S1) applied: “(…)that are at risk of becoming socially excluded”. How the author determined that the young people are being at risk of social exclusion? The NEET status does not always determine it. Please see: https://doi.org/10.1080/13676260600805671
Also, the timing and place of the data analyses is not mentioned.
7. Results
The descriptive results are integrated with all the analyses carried out.
8. Discussion
The key findings of the discussion are presented. It also includes the main strengths and weaknesses in relation to other studies, discussing important differences in the results.
Limitations should be moved to discussion.
9. References
The references are used correctly although there are many new references that could be used.
10. Tables and figures
Table S1 – the definition of NEET must be improved
Tables – please provide the exact indicator or indicators used for gathering the data on the NEET rate. Under the tables define “t (df)”.
Reviewer 2 Report
This report includes original and significant contribution to the literature.
Author(s) conducted a secondary analysis of data extracted from the Organisation for Economic Cooperation and Development database. The overarching goal was to examine determinants of adolescents' withdrawal from labour market and education.
The results revealed associations between some environment unfavorable conditions and adolescents' Not engaged in Employment, Education or Training (NEET) rate.
The work has some areas for improvement:
(1) The study’s aim needs to be better contextualized with respect to previous and present relevant research findings. In other words, the gap in the literature and the significance of the current contribution needs to be better highlighted in the introduction and/or discussion sections.
(2) The discussion section is extremely limited. Most of the discussion goes around the idea that the current findings are in line with previous ones, which relatively weakens the significance of the study. More attention needs to be given to what this study adds to the literature. In addition, author(s) concluded their work with a very general statement on the need for policy change. This part needs to be clearly linked to and supported by the findings.
